# Satiety of Edible Insect-Based Food Products as a Component of Body Weight Control

**DOI:** 10.3390/nu14102147

**Published:** 2022-05-21

**Authors:** Magdalena Skotnicka, Aleksandra Mazurek, Kaja Karwowska, Marcin Folwarski

**Affiliations:** 1Departament of Commodity Science, Medical University of Gdańsk, 80-211 Gdańsk, Poland; aleksandra.mazurek@gumed.edu.pl (A.M.); kaja.karwowska@gumed.edu.pl (K.K.); 2Departament of Clinical Nutrition and Dietetics, Medical University of Gdańsk, 80-211 Gdańsk, Poland; marcin.folwarski@gumed.edu.pl

**Keywords:** edible insects, hunger, satiety, proteins, *Acheta domesticus*, *Alphitobius diaperinus*, *Tenebrio molitor*

## Abstract

Among the many aspects determining the nutritional potential of insect-based foods, research into the satiating potential of foods is an important starting point in the design of new functional foods, including those based on edible insects. The aim of this study was to assess the satiating value of products with the addition of freeze-dried insect flour. The test material included wheat pancakes in which corresponding proportions of wheat flour were substituted with 10% Mw, 0% Mw, and 30% Mw of flour from freeze-dried *Tenebrio molitor*, 10% Bw, 20% Bw, and 30% Bw of flour from *Alphitobius diaperinus*, and 10% Cr, 20% Cr, and 30% Cr of flour from *Acheta domesticus*. The study included the characterisation of physico-chemical properties and their effect on the satiating potential of the analysed pancakes. A total of 71 healthy volunteers (*n* = 39 women, *n* = 32 men) with no food phobias were qualified for the study. Each subject rated the level of hunger and satiety before and after ingestion at 30 min intervals over the subsequent 180 min on two separate graphical scales. The rating was done on an unstructured 100 mm visual analogue scale (VAS). A portion intended for testing had a value of 240 kcal. The highest average satiety values were noted for the pancakes with an addition of 30% *Alphitobius diaperinus* (Bw) and with the addition of 20% and 30% addition of *Acheta domesticus* flour (Cr). The *Tenebrio molitor*-based products were the least satiating. However, the largest addition of 30% of an insect flour for each variant considerably increased the satiating potential as compared to the control sample. Satiety was influenced the most by the protein content in the test wheat pancakes. The results support the idea of a possible usage of insect-based food products in the composition of obesity treatment diets, carbohydrate-limiting diets, and as alternative sources of protein.

## 1. Introduction

The alarming rise in the indices reporting a global epidemic of overweight and obesity affects not only human health and quality of life but also has numerous environmental and economic implications. Many people live in an environment that promotes excessive consumption, which is a consequence of profound social, economic, and cultural changes. In the past few years, the international food market has noted an increase in the sales of functional products [1,2] as well as products characterised by high satiating potential, whose main task is to effectively suppress hunger [3,4,5]

On the consumer side, the increase in interest in highly satiating foods is a result of changes occurring in the area of expectations and requirements as regards the product and its characteristics. 

Having considered many proposals for the use of different food types, the use of edible insects in the diet appears to be worth analysing: on the one hand, as a potential source of high-quality protein [6,7,8], while, on the other, as a component for composing carbohydrate-limiting diets [9,10].

Insect proteins have a very high digestibility (76–98%) and contain all the essential amino acids. Insects are also rich in fat, contain essential mono- and polyunsaturated fatty acids, in particular, linoleic and linolenic acids. At the same time, they contain small amounts of polysaccharides which may lead to a feeling of satiety for a sufficiently long period of time and an extended time until the next meal. This fact was already observed in the production of animal feed. This was the impetus to consider the use of edible insects in human nutrition [11,12,13].

The use of insect enriched food with high-quality protein, but also the use of insects has an economic and ecological aspect. Thanks to the industrial farming of insects, it is possible to compose diets with lower costs, because insect proteins are cheaper than those from conventional meat production [7].

Apart from hormonal regulation of food intake, the feeling of satiety is influenced by physical and chemical characteristics of foods, in the context of both short-term satiation which is activated after food intake thanks to mechanoreceptors and long-term satiation based on the maintenance of the body’s energy balance according to the glucostatic, lipostatic and aminostatic theories [14,15].

Of all the food components, the strongest satiating properties are exhibited by protein [16,17] and dietary fibre [18,19]. Protein is an essential component of every diet. It is believed that an increased feeling of satiety results from elevated anorexigenic hormone levels, lowered orexigenic hormone levels, increased diet-induced thermogenesis, elevated plasma amino acid levels, increased gluconeogenesis in the liver and increased ketogenesis due to higher protein intake. The intake of amino acids induces complex signals located in the intestines, stomach, brain, and blood. Protein is known to increase energy expenditure due to much greater thermogenesis than carbohydrates and fat and an increase in protein intake protects resting energy expenditure (REE), thus preventing a decrease in fat-free mass (FFM) [20]. The role of protein continues to be an important aspect of analyses and research into the satiety of individual ingredients in foods. According to studies conducted by many authors, protein has the highest satiating potential compared to carbohydrates or fats [21,22,23]. The current data suggest that an increase in protein ingestion plays a crucial role in an increase in satiety. It is assumed that protein releases substances of a neurochemical nature, thus suppressing the appetite. In addition, high dietary protein intake reduces the synthesis of ghrelin [24] while stimulating the YY and GLP-1 peptide levels [25]. 

The above-mentioned role of protein prompted the authors to investigate the satiating potential of three insect-based flours. In order to introduce insects as a food in the West, it is important to design appropriate products to gain consumer acceptance in the first place. Research results show that, from the consumer’s perspective, insects can be introduced into foods most effectively in the form of a powder or flour. Recently, many studies have been focusing on designing products containing insects and examining their properties. The most commonly enriched products include bakery products [26,27,28,29,30,31], pasta [32,33,34,35], meat products, including burgers [36,37,38], and sweets [39,40,41,42]. 

In contrast, there are no studies into the possibility of using insects as products with high satiating potential. For this reason, the aim of the study was to assess the satiating value of products with the addition of freeze-dried insect flours. We incorporated flours from three different insects; mealworm, the buffalo worm and the cricket into wheat pancakes and investigated the chemical parameters influencing the hunger and satiety levels. Due to the inconclusive results of studies by other authors [43,44,45], it was assumed that determined levels of hunger and satiety after consuming the test samples can vary by gender. Consequently, the results were compared for men and woman.

## 2. Materials and Methods

### 2.1. Materials

Flours from the following freeze-dried edible insects were used in the study: the mealworm *(Tenebrio molitor)* (larvae), the buffalo worm *Alphitobius diaperinus* (larvae) and the cricket *Acheta domesticus* (the imago). The freeze-dried insects were acquired from a breeding facility in the Netherlands (Insecten kwekrij van de Ven Fortweg, Deurne, The Netherlands). The insects were then ground in a laboratory mill (IKA, A11 basic, Staufen, Germany) and the obtained flour was passed through a sieve.

### 2.2. Pancake Preparation

The pancakes were prepared in three variants that differed in the proportion of insect flours added (10%, 20%, 30%) by substituting the equivalent quantity of wheat flour for each of the three insect species being studied. The control sample was prepared with no insects added. The recipe for the pancakes was based on the basic formula and basic ingredients (wheat flour 450 type, milk with 2% fat content, rapeseed oil, eggs, and salt). The detailed composition and proportions are presented in Table 1.

The dough was pan-fried into small pancakes, which were then served to the subjects for testing in portions with a value of 240 kcal [46,47].

### 2.3. Nutrient Composition

The study determined the composition of insect flours. The following parameters of the flours from edible insects were analysed: ash content (AOAC 923.03), protein content (AOAC 950.36) (the protein content was calculated by applying the conversion coefficient of 6.25); crude fat content (AOAC 935.38), water content (AOAC 925.10), and total dietary fibre content (AOAC 991.43) (AOAC, 2006) in three replications.

The pancakes with the addition of insect flours were analysed, in relation to the control sample, in terms of moisture, ash, fat, and protein contents, using standard analytical methods (AOAC, 2005) and taking into account the correct conversion coefficient for insect protein. Carbohydrates were calculated based on the following formula: 100—(weight in grams (protein + fat + ash + moisture) in 100 g) [48].

The protein contents in the test insect flours were determined at 45.1 g × 100 g^−1^ for the mealworm, 58.6 g × 100 g^−1^ for the buffalo worm, and 69.1 g × 100 g^−1^ for the cricket. Therefore, the enrichment of pancakes with insect flour resulted in the expected proportional increase in the protein content in the test samples with an increase in the percentage of flour with the addition of three insect species (Table 2). A higher protein content in the cricket flour resulted in a greater increase in the protein content in finished pancakes as compared to the control sample, mealworm flour, and buffalo worm flour. Moreover, an increase in the protein content in the pancakes resulted in a reduced carbohydrate content (*p* < 0.05).

The fat content in the analysed flours varied significantly: 27.3 g × 100 g^−1^ in mealworm flour, 16.52 g × 100 g^−1^ in buffalo worm flour, and 10.71 g × 100 g^−1^ in *Acheta cricket* flour, which is reflected in the fat content in the prepared pancakes. The weight and volume of the samples intended for testing were very similar to each other, and the differences in these parameters were not statistically significant and had no effect on the determination of hunger and satiety levels. In all finished products, the protein content was higher than that in the control sample. The carbohydrate content decreased proportionally as well. The substitution of 30% wheat flour with cricket flour decreased the carbohydrate content by 45% as compared to the control sample. The fibre content changed as well. The higher the proportion of insect flour, the higher the fibre content in the end product was. The largest increases in the fibre content were noted for pancakes with the addition of *A. domesticus* flour (17–50%), while the smallest increases were for the buffalo worm (8–18%).

### 2.4. Satiety and Hunger Measurement

Satiating value of foods involves the investigation of appetite and tastiness. From this perspective, a subjective appetite is characterised by parameters such as satiety, hunger, the sensation of feeling full, expected food ingestion, thirst, and mood. The study regarded the determination of hunger and satiety levels as a reference method that indicated the relationships between the nutritional properties of individual samples and the hunger and satiety levels. For this reason, the assessment of tastiness and other parameters taken into account, e.g., psychological aspects and dietary habits, was abandoned. Studies based on an extended analysis of the appetite and tastiness most commonly concern characteristics at an ad libitum intake [49,50]. In the proposed research model, each subject rated the level of hunger and satiety they felt before and after the ingestion of the product at 30 min intervals over the subsequent 180 min on two separate graphical scales. The hunger and satiety levels were determined in the morning hours, between 08:00 a.m. and 09:00 a.m., and the subjects were fasting at the beginning of the study. The study was conducted for 4 months, from October 2021 to January 2022. Every research participant tested every sample every second day, but in random order. The research was a double-blind trial. The consumers participating in the study were of normal body weight, exhibited moderate physical activity, were using no medications and reported no diet-related diseases. The participants were not financially reimbursed for their time. 

The rating was done on an unstructured 100 mm visual analogue scale (VAS) with boundary markings to indicate satiety levels: 0 mm—“not satiated at all” and 100 mm—“very satiated”; 0 mm—“not hungry” and 100 mm—“very hungry”, to indicate the hunger level [51,52]. The subjects ingested the entire sample, with the time of ingestion being as short as possible and not exceeding five minutes. A portion intended for testing had a value of 240 kcal (1000 kJ). The rating on the VAS scale was done for 180 min, in the morning hours. The first measurement was taken on an empty stomach, while the next measurement was taken after the ingestion of a 240 kcal portion of the pancakes in different variants. Subsequent measurements were taken at 30 min intervals, except the first measurement after ingestion, which was taken after the first hour of testing, then after 60, 90, 120, and 150 min, and the last measurement was taken after 180 min. The results were presented using graphs by determining the area under the curve (AUC) using the trapezoidal method.

### 2.5. Consumer Characteristics

A total of 74 subjects participated in the experiment, including 41 women and 33 men. The eligibility criterion for the study was as follows:-BMI: from 18.5 to 25 kg/m^2^,-age from 20 to 28 years.

The study participants reported no chronic medical conditions, taking no medication or supplements and followed no special diet. All subjects signed a voluntary consent for participation in the study that was approved by the Independent Institutional Ethics Committee for Scientific Research at the Medical University of Gdańsk (NKBBN/349/2021). The study participants completed a questionnaire containing questions about age, body weight, height, cigarette smoking, and physical activity. Four people who were not able to complete the experiment were excluded from the final analysis. Ultimately, 71 subjects were qualified for the study.

The first stage was to designate a representative group for the study. The test power analysis and interval estimation concerning the sample size were performed using the STATISTICA 13.0 package. The analysis enabled, as early as the experiment planning stage, the selection of the sample size at a level that provides statistical conclusions with adequate accuracy, confidence, and probability. Statistical analysis demonstrated that all of the tests used to characterise changes in the level of hunger felt, conducted for nine different product variants, can be regarded as representative. Based on the collected data concerning the assessment of hunger and satiety, the minimum sample size for each sample should be 21 for all the measurements, regardless of gender. In contrast, for the separate samples of women and men, it was required to perform at least 19 measurements. A Q-Dixon test was also performed to reject possible initial errors. However, each time, the calculated value of Q was lower than the value determined from the Q_kr_ tables, which indicated that the doubtful result was part of the sample. 

The study involved the investigation of changes in the level of subjectively felt hunger and satiety associated with the ingestion of selected types of pancakes with the addition of edible insect flour and the determination of the effect of selected chemical parameters of the analysed products on these phenomena. The study was designed to reduce the effect of external elements that could interfere with the results. Every study participant was trained on the meaning of the concepts of hunger and satiety. It was emphasised that the flavour, preferences, habits, and the way the test samples were served did not determine the actual levels of hunger and satiety felt at any stage of the study. The only factor to exclude the consumer from the assessment was food aversion and/or allergy to insects. The experiment was carried out in the morning hours, and the last ingested meal was supper on the previous day. 

### 2.6. Statistical Analysis

All statistical analyses were conducted using a statistical package by StatSoft Inc. (Cracow, Poland) (2014), Statistica (data analysis software system), version 12.0. and an Excel spreadsheet. 

At the experiment planning stage, the selection of the sample size at a level that provides statistical conclusions with adequate accuracy and confidence (and the probability of the detection of the effects of the given size by the test) were investigated based on the test power analysis and interval estimation concerning the sample size. Quantitative variables were characterised using the arithmetic mean and standard deviation (SD), while qualitative variables were presented using the sizes and percentages.

The significance of differences between women and men was checked using the one-way ANOVA variants with repeatable parameters. 

The parameters of multiple regression taking into account the concept of shared variability were estimated using the REGLINP command in an Excel 2010 PL spreadsheet.

## 3. Results

### 3.1. Hunger

The level of hunger felt in the entire study population was VAS = 71.1; SD (6.8). Women rated the subjective feeling of hunger before the ingestion of meal slightly lower (VAS = 70.7); SD (7.1) in relation to the level of hunger felt by men, VAS = 72.1; SD (6.6). The ANOVA analysis of variance with repeated measures found no statistically significant differences (*p* = 0.0855) in the group of women in the assessment of hunger on an empty stomach. Similarly, in the group of men, no statistically significant changes (*p* = 0.1285) were noted in the assessment of the hunger level on an empty stomach. The analysis of variance with repeated measures demonstrated that for eight samples: control (*p* = 0.4923), 10% Mw (*p* = 0.2688), 10% Cr (*p* = 0.0832), 30% Cr (*p* = 0.2000), 30% Mw (*p* = 0.1003), 20% Cr (*p* = 0.3336), 20% Mw (*p* = 0.0568), and 10% Bw (*p* = 0.9010), no statistically significant differences were noted in the assessment of the feeling of hunger related to their ingestion, and determined by the gender of the study participants. Regarding these variants, the hypothesis assuming an effect of the gender on the level of hunger felt after their ingestion was rejected. For the two remaining food samples under analysis, the ANOVA test value demonstrated statistically significant differences in the assessment of the feeling of hunger determined by gender. Regarding the ingestion of 30% Bw (*p* = 0.0101) and 20% Bw (*p* = 0.0469), the level of hunger felt by the women received a higher mean score as compared to the men, with the differences being statistically significant at a significance level of α = 0.05.

Overall, it can be concluded that the hunger level felt at subsequent time intervals after the ingestion of isocaloric test samples in the majority of episodes was not determined by gender. The analysis of available data in terms of the effect of the gender on the hunger level, using graphical scales, indicated no relationship between the feeling of the hunger level among the women and men, which was confirmed by studies presented in this paper [44,45]. For this reason, further discussion of the results will be held for the entire population.

Immediately after ingestion, the samples with the addition of *A. diaperinus* larvae flour minimised the feeling of hunger the best. The VAS value in these cases was similar to that for the feeling of hunger after the ingestion of the control sample, i.e., VAS_c_ = 17.6 (6.1). Having analysed the entire period of the testing, i.e., 180 min, it must be stated that of all the test variants, which were characterised by high hunger-suppressing potential immediately after ingestion, all samples, excluding one with the addition of 30% of *A. diaperinus* (Bw) flour, rapidly initiated the occurrence of increased hunger during the later part of the study. The ingestion of pancakes with the largest proportion (30% Bw) of the buffalo worm determined the feeling of a low hunger level not only immediately after its ingestion but also during the subsequent measurements. The feeling of hunger returned most rapidly after the ingestion of 10% Mw, 10% Bw, and the control sample. After three hours the noted values of hunger were VAS_10%Mw_ = 76.6 (8.1), VAS_10%Bw_ = 76.0 (8.2), and VAS_c_ = 74.5 (8.3), respectively. The products which, in the subjects’ opinion, suppressed the feeling of hunger the best (rapidly and for a long time) were the samples with the addition of *A. domesticus* flour (20% and 30% Cr) (Table 3).

Studies using graphical methods refer to individual measurements that are limited by a subjective error of consumers. Therefore, in order to quantify primary subjective data, the study estimated the AUC values of the hunger level [53], which enabled the conclusion that the lowest hunger level was noted for the ingestion of samples with the addition of 20% and 30% *A. domesticus* (Cr) flour and with the addition of 30% *A. diaperinus* (Bw) flour (Table 4). Statistical analysis using the ANOVA test showed that the hunger index was determined by the species of the insect and the percentage of its addition (α = 0.05, F = 71.22 of the critical value).

(Figure 1) presents changes in the hunger level after the ingestion of test samples. Graphical interpretation of the obtained results shows that the most critical jump in the feeling of hunger was noted after 90 min from the ingestion. After that time, a steadily increasing level of hunger felt was observed.

### 3.2. Satiety

Hunger and satiety are two major feelings that control food intake. Similar to the study into the hunger level, an analysis of the feeling of satiety was also conducted. The level of satiety felt on an empty stomach in the entire study population was VAS = 28.1; SD (9.8). The study was carried out in parallel with the measurement of the satiety level in the morning hours, and the last ingested meal was supper on the previous day. Immediately after ingestion, the samples with the addition of *A. diaperinus* (Bw) proved to be the most satiating. The satiety value was similar to the satiety level for the control sample VAS_c_ = 81.2; SD (9.2). 

The statistical analysis demonstrated that, for seven samples, no statistically significant differences, determined by the gender of the study participants, were noted in the assessment of satiety. For the control sample (*p* = 0.4157), 10% Mw (*p* = 0.4157), 30% Mw (*p* = 0.1312), 10% Cr (*p* = 0.3307), 20% Cr (*p* = 0.1283), 30% Cr (*p* = 0.8162), and 10% Bw (*p* = 0.8877), gender had no effect on the level of satiety of test pancakes. For three variants, there were statistically significant differences in the assessment of satiety determined by the gender of the study participants. Regarding the ingestion of 20% and 30% Bw, and 20% Mw, the women felt a higher satiety level compared to the men (*p* = 0.0081 for 30% Bw, *p* = 0.0081 for 20% Bw and *p* = 0.0014 for 20% Mw). The asymmetry of the results concerning the rating of the level of satiety felt, which was identified for three samples, similar to the study into the level of felt hunger, could have been due to the same interference. 

The obtained results indicate that all the test products with the addition of insect flour were characterised by a high capacity to induce the feeling of satiety immediately after ingestion. For every test sample, the determined value exceeded VAS = 75.2 (Table 5). 

The determined level of the satiety felt in successive time measurements showed certain variability. Each of the test samples was characterised by a relatively high capacity to induce the feeling of satiety, which steadily decreased. However, the ingestion of different variants was accompanied by varying intensity of changes in the feeling of satiety over time. The results obtained at the last point of the measurements of changes in satiety over time suggested that the products which (according to the study participants) maintained the feeling of satiety the best were the samples with the addition of 20% and 30% of the *A. domesticus* (Cr) and 30% of *A. diaperinus* (Bw). For each of the products under study, the level indicated on the graphical scale after 180 min exceeded the VAS value of 45, which indicated their good satiating properties. The lowest satiating value was identified for 10% Mw VAS = 22.3 SD (9.0) and 10% Bw VAS = 21.4 SD (8.0). Following the entire test, after 180 min, the satiety level was at a level similar to that of the control sample with no insect flour added. This means that the ingestion of these pancakes with a small (only 10%) addition of flour from mealworm and buffalo worm resulted in a rapid yet short-lived increase in the level of satiety felt, which just as rapidly gave way to the increasing feeling of hunger.

Similar to the first part of the result analysis, in order to describe changes in the satiety label over time, the AUC values, determined by the trapezoidal method, were used (Table 4). A comparison of the obtained results enabled the selection of samples characterised by the largest area under the curve (which indicated their higher total satiating capacity) while allowing the changes in the level of satiety felt over time to be analysed. The largest area was determined for the samples 30% Bw, 20% Cr, and 30% Cr, which were thus recognised as the most satiating samples (Figure 2). A statistical analysis of the results concerning the presented stage demonstrated that the satiating capacity is determined by the sample type (α = 0.05, F = 65.24 of the critical value). Similar results were obtained from a study conducted on a group of women (F = 62.26) and men (F = 71.35).

### 3.3. The Effect of Physico-Chemical Factors on the Generation of Satiety in Products with the Addition of Insect Flour

The application of graphical methods in the research into the feeling of hunger and satiety after the ingestion of isocaloric portions of different types of samples with the addition of an insect flour helped determine the critical parameters that describe the satiating value of the test variants. A comparative analysis of the results obtained by means of estimating the area under the curve (AUC) enabled the conclusion that the level of satiety felt after the ingestion of test pancakes was a complementary quantity in relation to the level of hunger felt after their ingestion. For this reason, in order to determine the physico-chemical characteristic relationships, only the results of the satiety level determination were used, and the obtained implications were presented in the form of a multiple regression analysis. 

In order to determine which of the factors being considered had the strongest effect on the feeling of satiety, Pearson correlation coefficient values were determined for individual variables. Having analysed the relationship between the AUC satiety levels determined for all products and the physico-chemical parameters, it was found that the highest parameter values were identified for protein (0.82) and dietary fibre (0.58). On the other hand, the contents of water (−0.63) and carbohydrates (−0.68) were inversely proportional to the satiety level. The fitting of these characteristics was not strong, which supports the thesis about the complexity of the phenomenon under study. By applying the previously used approach, which assumes that the simultaneous use of many independent variables will serve to increase the prediction accuracy, the study estimated the parameters of the equation describing the effect of the chemical agent content in the test samples on the level of satiety felt after their ingestion, expressed in the satiety AUC values. 

The equation was determined based on chemical data. The obtained multiple equation of satiety took the following form: 
y = 9.82·x_1_ + 1.04·x_2_ − 26.17·x_3_ + 2.30·x_4_ − 5.46·x_5_ + 267.56
(1)

where:
y—AUC M satiety, x_1_—protein content, x_2_—carbohydrate content, x_3_—fibre content, x_4_—fat content,x_5_—water content.


The value of the squared correlation coefficient R^2^ was 0.95, which means that it expresses the dominant part of the variance shared by the best combination and the dependent variable. Mathematical model proved to be statistically significant and all predictors explained 95% of dependant values. The obtained results indicate that the level of satiety felt was a resultant of the predictors taken into account in the regression equation, of which the protein content was the most important for inducing the feeling of satiety. The water content and content of fibre were of the smallest importance for inducing the feeling of satiety, which is evidenced by the negative value of this parameter (counteracting the phenomenon under study). The other variables had positive values in the determined equation, which suggests that their presence supported the promotion of the feeling of satiety (interaction with the phenomenon under study). 

## 4. Discussion

To our knowledge, this is the first study to determine the satiety of insect-derived sources for protein in food. The determination of the satiating capacity of food products is extremely complicated because in carrying out the study, one has to deal with the independently occurring determinants concerning the physiological nature of the phenomenon under study and the cognitive problems related to the development of relevant research methods. 

In diet therapy and the treatment of overweight, the meals should be composed in such a way that they can quickly ensure the feeling of fulness in the stomach and maintain the state of satiety for as long as possible. In the proposed experiment, immediately after ingestion, the samples with the addition of *A. diaperinus* (Bw), which corresponded to the defined first variant, were the most satiating. A portion of 240 kcal, large in volume, caused a temporary feeling of fulness. Only the sample with the addition of 30% Bw exhibited high satiating potential, both immediately after ingestion and after 180 min. The area under the curve (AUC) determined for this sample was the largest and amounted to 196.55. Such a high value of the area under the curve was determined by the high values noted for all seven measurements taken over time. At the same time, it should be stressed that, having compared the study results concerning the feeling of satiety after 180 min from the ingestion of all test samples, the highest satiety level was noted for the samples with the addition of 10%, 20%, and 30% Cr, which thus represented the third variant. The study results suggest that the products richer in protein (30% Bw, 30% Cr) suppressed hunger and promoted satiety much more strongly, and the multiple equation determined for satiety confirmed the complexity of the phenomenon. Moreover, it was determined that the feeling of satiety was contributed to the most in all the analysed predictors in this model terms by the protein content in test wheat pancakes. In addition, the statistical analysis of the study results showed that the protein content varied significantly depending on the sample type (*p* = 0.004).

Understanding the protein digestion process is difficult to interpret. It is suggested that a crucial role is played by the levels of CCK whose activity is stimulated at different stages of digestion. This substance stimulates the motor activity of the intestines and slows down the process of gastric emptying [54]. In order to raise the satiety level, research is being conducted into the enrichment of foods with a variety of proteins [55,56]. Indeed, the results of these findings enable an optimistic assessment of the beneficial function of protein on influencing satiety. 

As regards the diets with reduced carbohydrate and increased protein supply, the process of protein-mediated gluconeogenesis can occur. In this situation, the reaction that modulates satiety is probably not related to the involvement of protein but rather to the increased gluconeogenesis [57]. It can be concluded that a high-protein meal increases the likelihood of a reduction in the energy content of the subsequent meal by as much as 25%. Furthermore, a reduction in the carbohydrate content can also have a positive effect on influencing satiety, as carbohydrates are mentioned as components with the lowest satiating potential [58,59]. The interpretation of measurements using the visual analogue scale (VAS) is very difficult, as it does not enable an unambiguous conclusion as to what score obtained using it can be regarded as high in the assessment of satiety, and what score can only be regarded as satisfactory. For this reason, the VAS-based consumer assessment should be regarded as a reference method that can facilitate the result interpretation based on physico-chemical parameters. Previous studies highlighted the beneficial role of protein in promoting satiety. The satiating properties of dairy products [60,61] and animal protein [62,63] were particularly highly rated. Our study, based on the concept of enriching foods with insect protein, is consistent with this trend. Research into the satiety potential of insects is a new but extremely topical issue in terms of its nutritional, ecological and economic aspects. The originality of the conducted study is proven by the fact that an attempt was made for the first time to determine the satiating properties of cereal products with the addition of flours from three insect species based on a subjective consumer assessment using the VAS scale, while taking into account the chemical composition of the proposed test variants. Our findings suggest that there are differences in the feeling of hunger and satiety after the ingestion of different insects and their amounts in the test samples. A limitation of the study is the small meal (~240 kcal) used in the experiment that is equivalent to a snack rather than a full meal. It should be emphasized, however, that this was the first study on the subject, and the simplicity of the test sample was intended to draw clear conclusions about the saturation of well-being from purely insect-derived protein sources. Our conclusions could lead to future research with more complex nutritional patterns based on the above-mentioned ingredients. 

Research into the satiety provided by insects should be one of the directions indicating the possibilities for their use. The knowledge of these characteristics could help compose protein-rich diets and design new functional foods using insects, but also in the treatment of obesity.

## 5. Conclusions

The study into changes in the hunger and satiety levels determined by the ingestion of test food products with the addition of different proportions of insect flours, assessed their satiating value, and the physico-chemical assays identified the main determinants of the satiating potential of the proposed products. The study results confirmed that the level of satiety felt after ingestion varied depending on the amount of insect flour added and on the insect species. The most satiating pancakes were those with the addition of 30% *Alphitobius diaperinus* flour (Bw) and with the addition of 20% and 30% *Acheta domesticus* flour (Cr). There is a strong influence of physico-chemical characteristics on the induction of satiety, of which the protein content proved to be the most significant parameter.

## Figures and Tables

**Figure 1 nutrients-14-02147-f001:**
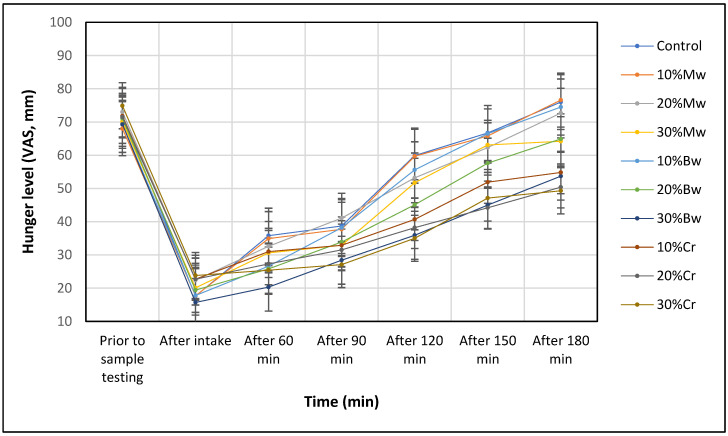
The determined hunger level after the ingestion of samples with the addition of edible insects for the entire study population.

**Figure 2 nutrients-14-02147-f002:**
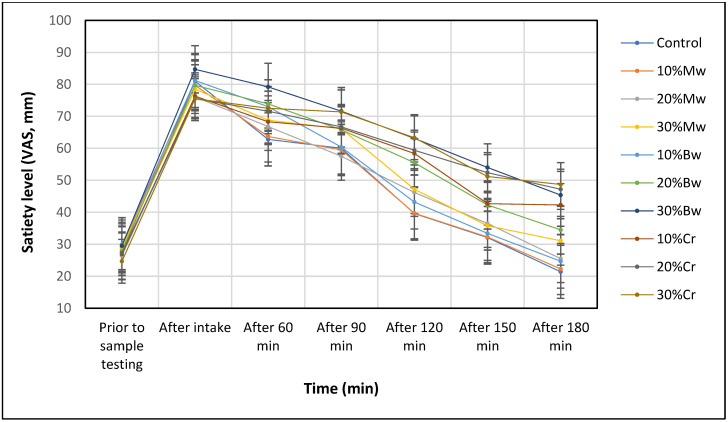
The determined satiety level after the ingestion of samples with the addition of edible insects for the entire study population.

**Table 1 nutrients-14-02147-t001:** Nutritional value of freeze-dried edible insects (g × 100 g^−1^).

Insect Species	Protein	Fat	Ash	Fibre	Energy kcal/kJ/100 g
Tenebrio molitor	45.31 ± 0.02 ^a^	27.31 ± 0.45 ^c^	1.65 ± 0.05 ^b^	8.52 ± 0.05 ^a^	550 ± 2.3
Alphitobius diaperinus	58.62 ± 0.01 ^b^	16.52 ± 0.08 ^a^	1.54 ± 0.05 ^b^	6.24 ± 0.07 ^c^	484 ± 3.0
Acheta domesticus	69.17 ± 0.05 ^c^	10.71 ± 0.32 ^b^	2.78 ± 0.05 ^b^	9.75 ± 0.11 ^b^	458 ± 2.1

Values with different letters in the same column indicate significant differences (*p* < 0.05).

**Table 2 nutrients-14-02147-t002:** Nutritional value of a pancake portion (240 kcal).

Sample		Weight (g)	Protein (%)	Fat (%)	Carbo- hydrates (%)	Ash (%)	Fibre (%)	Moisture (%)
Control	C	94	8.13 ± 0.19 ^b^	7.93 ± 0.1 ^ac^	34.23 ± 0.88 ^a^	0.98 ± 0.05 ^c^	1.38 ± 0.06 ^a^	40.34 ± 1.23 ^a^
Tenebrio molitor larvae	10% Mw	91	9.29 ± 0.54 ^b^	9.15 ± 0.12 ^c^	29.39 ± 0.76 ^a^	0.96 ± 0.01 ^c^	2.03 ± 0.01 ^b^	40.69 ± 0.98 ^b^
	20% Mw	89	10.53 ± 0.68 ^b^	10.84 ± 0.52 ^d^	24.33 ± 0.45 ^d^	0.96 ± 0.01 ^c^	2.54 ± 0.01 ^b^	42.96 ± 0.97 ^b^
	30% Mw	87	11.73 ± 0.38 ^ac^	12.37 ± 0.22 ^d^	19.65 ± 0.66 ^bc^	0.97 ± 0.05 ^c^	3.10 ± 0.04 ^ac^	41.46 ± 1.01 ^ac^
Alphitobius diaperinus larvae	10% Bw	92	9.98 ± 0.16 ^b^	8.70 ± 0.37 ^b^	29.73 ± 0.43 ^a^	0.91 ± 0.02 ^c^	1.96 ± 0.05 ^b^	40.75 ± 1.28 ^a^
	20% Bw	92	11.99 ± 0.85 ^a^	9.96 ± 0.45 ^ad^	25.05 ± 0.33 ^d^	0.87 ± 0.04 ^c^	2.40 ± 0.05 ^bc^	40.66 ± 1.22 ^a^
	30% Bw	90	14.00 ± 0.29 ^abc^	11.10 ± 0.10 ^ab^	20.55 ± 0.77 ^b^	0.86 ± 0.04 ^c^	2.89 ± 0.04 ^ac^	40.81 ± 0.65 ^bc^
Acheta domesticus imago	10% Cr	92	10.48 ± 0.44 ^a^	8.50 ± 0.19 ^b^	29.64 ± 0.55 ^a^	0.99 ± 0.05 ^c^	2.12 ± 0.01 ^de^	39.06 ± 0.45 ^c^
	20% Cr	92	13.05 ± 0.34 ^c^	9.54 ± 0.29 ^cd^	24.69 ± 0.88 ^d^	1.12 ± 0.04 ^c^	2.72 ± 0.01 ^e^	40.54 ± 0.67 ^ac^
	30% Cr	92	15.66 ± 0.66 ^c^	10.46 ± 0.18 ^bc^	19.94 ± 0.26 ^c^	1.21 ± 0.06 ^c^	3.37 ± 0.08 ^d^	42.93 ± 0.98 ^ab^

Values with different letters in the same column indicate significant differences (*p* < 0.05).

**Table 3 nutrients-14-02147-t003:** The results of the hunger level assessment after the ingestion of test products with the addition of an insect flour for *n* = 71 participants.

**Item**		Control		
Before ingestion	mean (SD)	71.8 (8.3)		
After ingestion	mean (SD)	17.6 (6.1)		
After 180 min	mean (SD)	76.0 (8.2)		
**Item**		Tenebrio molitor 10%	Tenebrio molitor 20%	Tenebrio molitor 30%
Before ingestion	mean (SD)	68.0 (9.9)	73.0 (6.6)	70.1 (10.2)
After ingestion	mean (SD)	17.7 (5.8)	22.4 (6.8)	22.2 (8.8)
After 180 min	mean (SD)	76.6 (8.1)	72.7 (8.8)	64.2 (9.3)
**Item**		Alphitobius diaperinus 10%	Alphitobius diaperinus 20%	Alphitobius diaperinus 30%
Before ingestion	mean (SD)	69.3 (9.8)	71.1 (8.2)	69.31 (0.8)
After ingestion	mean (SD)	17.8 (9.1)	19.4 (8.5)	15.7 (7.8)
After 180 min	mean (SD)	74.5 (8.3)	64.9 (9.3)	53.7 (9.7)
**Item**		Acheta domesticus 10%	Acheta domesticus 20%	Acheta domesticus 30%
Before ingestion	mean (SD)	71.8 (9.5)	71.5 (7.2)	74.9 (6.2)
After ingestion	mean (SD)	22.7 (8.2)	22.7 (6.0)	23.8 (5.8)
After 180 min	mean (SD)	54.8 (9.0)	50.4 (7.2)	49.3 (9.2)

**Table 4 nutrients-14-02147-t004:** The determined areas under the curve (AUC) for the assessment of hunger and satiety in test samples.

Sample	Hunger AUC (SD)	Satiety AUC (SD)
Control	149.89 (9.2) *	151.32 (8.2) *
10% Mw	147.33 (6.9) *	151.76 (9.5) *
20% Mw	146.07 (8.9) *	155.85 (7.7) *
30% Mw	136.28 (10.3)	165.12 (11.1)
10% Bw	141.74 (9.1) *	160.75 (5.6) *
20% Bw	128.46 (7.7)	175.52 (3.3)
30% Bw	106.87 (6.7) **	196.55 (8.9) **
10% Cr	124.94 (6.2)	176.90 (8.1)
20% Cr	116.00 (7.4) **	182.84 (6.7) **
30% Cr	114.00 (6.8) **	186.84 (6.50) **

*—the highest determined AUC value. **—the lowest determined AUC value.

**Table 5 nutrients-14-02147-t005:** The results of the satiety level assessment after the ingestion of food products with the addition of an insect flour for *n* = 71 participants.

**Item**		Control	
Before ingestion	mean (SD)	27.4 (8.20)	
After ingestion	mean (SD)	81.0 (5.6)	
After 180 min	mean (SD)	21.4 (8.00)	
**Item**		Tenebrio molitor 10%	Tenebrio molitor 20%
Before ingestion	mean (SD)	29.6 (9.9)	26.4 (6.5)
After ingestion	mean (SD)	79.7 (5.5)	76.1 (6.6)
After 180 min	mean (SD)	22.3 (9.0)	25.5 (8.1)
**Item**		Alphitobius diaperinus 10%	Alphitobius diaperinus 20%
Before ingestion	mean (SD)	29.9 (9.5)	27.9 (8.9)
After ingestion	mean (SD)	81.2 (9.2)	79.7 (7.7)
After 180 min	mean (SD)	24.7 (10.0)	34.2 (8.8)
**Item**		Acheta domesticus 10%	Acheta domesticus 20%
Before ingestion	mean (SD)	26.9 (8.8)	27.3 (6.9)
After ingestion	mean (SD)	76.3 (7.7)	75.6 (5.7)
After 180 min	mean (SD)	42.3 (8.7)	47.2 (5.3)

## Data Availability

Data are contained within this paper, and no further information about qualitative data can be shared due to ethical/privacy reasons as we worked with vulnerable communities.

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
