# Peer review of "Satiety of Edible Insect-Based Food Products as a Component of Body Weight Control"

_nutrients, 2022, doi:10.3390/nu14102147_

Round 1
Reviewer 1 Report
Overall the study has been not well performed. The English language needs to be corrected thoroughly.
The title, abstract and Discussion need improve
Rewrite the introduction, Materials and Methods, Results, Discussion, and Conclusion.
They also used the self citation
Skotnicka, M.; Karwowska, K.; KÅ‚obukowski, F.; Borkowska, A.; Pieszko, M. Possibilities of the Development of Edible Insect-658 Based Foods in Europe. Foods. 2021, 10(4), 766; https://doi.org/10.3390/foods10040766.
Kowalski, S.; Mikulec, A.; Mickowska, B.; Skotnicka, M.; Mazurek, A. Wheat bread supplementation with various edible insect 613 flours. Influence of chemical composition on nutritional and technological aspects. LWT, 2022, 159, 113220; 614 https://doi.org/10.1016/J.LWT.2022.113220.
Author Response
Dear Reviewer,
We are honored and grateful for reviewing our manuscript. The manuscript has been corrected according to all the suggestions and submitted in track-changes mode. All modifications are described in our reply below.
They also used the self citation
Skotnicka, M.; Karwowska, K.; KÅ‚obukowski, F.; Borkowska, A.; Pieszko, M. Possibilities of the Development of Edible Insect-658 Based Foods in Europe. Foods. 2021, 10(4), 766; https://doi.org/10.3390/foods10040766.
Kowalski, S.; Mikulec, A.; Mickowska, B.; Skotnicka, M.; Mazurek, A. Wheat bread supplementation with various edible insect 613 flours. Influence of chemical composition on nutritional and technological aspects. LWT, 2022, 159, 113220; 614 https://doi.org/10.1016/J.LWT.2022.113220.
Thank you very much for this remark. We agree, that extensive self-citation is not recommended, however scientific researches in this aera are extremely limited. Since our study is a consequence and continuity of previous finding we published, the aim of the citation is not to boost the statistics but just to point continuity and available literature.
We have added modifications in the study following the remarks of all the reviewers. Required English language corrections have been introduced following additional revision of native speaker.
Once again, we are very thankful for your review and contribution to the improvement of the study. We do hope that we have met all your expectations within resubmitted article.
Kind Regards
Magdalena Skotnicka, PhD

Reviewer 2 Report
111-113 The freeze-dried insects were acquired from a breeding facility in the Netherlands. The insects were then ground in a laboratory mill (IKA A11 basic) and the obtained flour was passed through a sieve. – please, identify the facility in the Netherlands and the manufacturer of the laboratory mill (including the country).
Figure 1, 2 - the values ​​in the graphs overlap, the graphs should be larger (higher), to make them better distinguishable.It would be a good idea to either add a guide grid or add y-axis labels to the other side.
It would be good to emphasize more why to use insects to increase protein content in reduction diets. - Quality protein? Cheaper? Modern? Tastier?
I would suggest not to use the same words in keywords as in the Title, it would increase a chance for the article to be found when searching through the scientific databases. I would recommend to add e.g. alphitobius diaperinus and other insect names.
Author Response
Dear Reviewer,
We are honored and grateful for reviewing our manuscript. All your suggestions and corrections are very valuable and helpful for us. The manuscript has been corrected according to all the suggestions and submitted in track-change mode. All modifications are described in our reply below.
111-113 The freeze-dried insects were acquired from a breeding facility in the Netherlands. The insects were then ground in a laboratory mill (IKA A11 basic) and the obtained flour was passed through a sieve. – please, identify the facility in the Netherlands and the manufacturer of the laboratory mill (including the country).
All required information have been updated in the text, including mill’s type and facility details.
Figure 1, 2 - the values ​​in the graphs overlap, the graphs should be larger (higher), to make them better distinguishable.It would be a good idea to either add a guide grid or add y-axis labels to the other side.
Graphs have been re-formatted to improve readability including auxiliary grids. The graphs size limitations disable increasing resolution.
It would be good to emphasize more why to use insects to increase protein content in reduction diets. - Quality protein? Cheaper? Modern? Tastier?
Thank you for that suggestion. In line [85],the additional excerpt on the purpose of using insect protein in diets has been included.
I would suggest not to use the same words in keywords as in the Title, it would increase a chance for the article to be found when searching through the scientific databases. I would recommend to add e.g. alphitobius diaperinus and other insect names.
Thank you very much for the valuable hint. The keywords have been updated with additional words.
Once again, we are very thankful for your review and contribution to the improvement of the study. We do hope that we have met all your expectations within resubmitted article.
Kind Regards
Magdalena Skotnicka PhD

Reviewer 3 Report
2.1 Materials
Precise scientific determination of species is not necessary in this case, as these insect species are well known in entomophagy and cannot be confused. If the authors still want to state the exact zoological classification (including taxonomic categories and author), it is necessary to proceed using the approved nomenclature, where parentheses and dates play an important role. E.g.: https://www.biolib.cz/en/taxon/id14741/ the mentioned species are named as follows:
Tenebrio molitor Linnaeus, 1758
Alphitobius diaperinus (Panzer, 1797)
Acheta domesticus (Linnaeus, 1758)
The parentheses, names and years must be used precisely; the use or non-use of parentheses changes the information about how the naming of the species has been used in history. In this case, I would suggest using only the Latin names for simplicity.
164 - age from 20 to 28 years - very narrow age group, can general conclusions be drawn accordingly?
Author Response
Dear Reviewer,
We are honored and grateful for reviewing our manuscript. All your suggestions and corrections were very valuable and helpful for us. The manuscript has been corrected according to all the suggestions and submitted in track-changes mode. All modifications are described in our reply below.
2.1 Materials
Precise scientific determination of species is not necessary in this case, as these insect species are well known in entomophagy and cannot be confused. If the authors still want to state the exact zoological classification (including taxonomic categories and author), it is necessary to proceed using the approved nomenclature, where parentheses and dates play an important role. E.g.: https://www.biolib.cz/en/taxon/id14741/ the mentioned species are named as follows:
Tenebrio molitor Linnaeus, 1758
Alphitobius diaperinus (Panzer, 1797)
Acheta domesticus (Linnaeus, 1758)
The parentheses, names and years must be used precisely; the use or non-use of parentheses changes the information about how the naming of the species has been used in history. In this case, I would suggest using only the Latin names for simplicity.
Thank you very much for this valuable comment. In order to systematize the nomenclature, we limited ourselves to Latin names and simplified descriptions (flour from individual insects), what is acceptable in this type of publication, because flour can be treated as a final commercial product.
164 - age from 20 to 28 years - very narrow age group, can general conclusions be drawn accordingly?
The group is narrow indeed, however we wanted it to be as homogeneous as possible in order to minimize possibility of any potential interferences. The group in the age range of 20-28 was selected because this is most likely target group for consumers willing to include insect products in the diet.
However, the current study is part of a larger project that takes into account the elderly persons as well as those affected by obesity, what will provide in the future a better and broader view of this new area of research.
Once again, we are very thankful for your review and contribution to the improvement of the study. We do hope that we have met all your expectations within resubmitted article.
Kind regards
Magdalena Skotnicka PhD

Round 2
Reviewer 1 Report
Overall the study has been not well performed. The English language needs to be corrected thoroughly.
The title, abstract and Discussion need improve.
Rewrite the introduction, Materials and Methods, Results, Discussion, and Conclusion.
They also used the self citation
Skotnicka, M.; Karwowska, K.; KÅ‚obukowski, F.; Borkowska, A.; Pieszko, M. Possibilities of the Development of Edible Insect-658 Based Foods in Europe. Foods. 2021, 10(4), 766; https://doi.org/10.3390/foods10040766.
Kowalski, S.; Mikulec, A.; Mickowska, B.; Skotnicka, M.; Mazurek, A. Wheat bread supplementation with various edible insect 613 flours. Influence of chemical composition on nutritional and technological aspects. LWT, 2022, 159, 113220; 614 https://doi.org/10.1016/J.LWT.2022.113220.
Why they did not use the updated study for example
- Gryllus testaceus walker (crickets) farming management, chemical composition, nutritive profile, and their effect on animal digestibility.
- Nutritional composition of various insects and potential uses as alternative protein sources in animal diets
Author Response
Dear Reviewer
We would like to thank you for consecutive review of our manuscript. We are grateful for your suggestions related to the improvement of our publication with additional references. The work is a clinical trial and is based on the use of insects in human nutrition and we have limited ourselves to that. However, it might be worth mentioning the use of insects for feeding animals as a valuable source of protein.
Therefore, below I present the list of publications that have been added to the manuscript.
- Shah AA, Wanapat M. Gryllus testaceus walker (crickets) farming management, chemical composition, nutritive profile, and their effect on animal digestibility. Entomol Res. 2021;51(12):639-649. doi:10.1111/1748-5967.12557
- Shah AA, Totakul P, Matra M, Cherdthong A, Hanboonsong Y, Wanapat M. Nutritional composition of various insects and potential uses as alternative protein sources in animal diets. Anim Biosci. 2022;35(2):317-331. doi:10.5713/ab.21.0447
- Hong J, Han T, Kim YY. Mealworm (Tenebrio molitor Larvae) as an Alternative Protein Source for Monogastric Animal: A Review. Animals. 2020;10(11):2068. doi:10.3390/ani10112068
Once again, we are very thankful for your review and contribution to the improvement of the study. We do hope that we have met all your expectations within resubmitted article.
Kind Regards
Magdalena Skotnicka PhD
